# Biclustering of Log Data: Insights from a Computer-Based Complex Problem Solving Assessment

**DOI:** 10.3390/jintelligence12010010

**Published:** 2024-01-17

**Authors:** Xin Xu, Susu Zhang, Jinxin Guo, Tao Xin

**Affiliations:** 1Collaborative Innovation Center of Assessment for Basic Education Quality, Beijing Normal University, Beijing 100875, China; xux636@nenu.edu.cn; 2Departments of Psychology and Statistics, University of Illinois Urbana-Champaign, Champaign, IL 61820, USA; szhan105@illinois.edu; 3College of Science, Minzu University of China, Beijing 100081, China; guojx@muc.edu.cn; 4School of Educational Science, Anhui Normal University, Wuhu 241000, China

**Keywords:** log file data, biclustering, PISA, action sequence, process data, timing data

## Abstract

Computer-based assessments provide the opportunity to collect a new source of behavioral data related to the problem-solving process, known as log file data. To understand the behavioral patterns that can be uncovered from these process data, many studies have employed clustering methods. In contrast to one-mode clustering algorithms, this study utilized biclustering methods, enabling simultaneous classification of test takers and features extracted from log files. By applying the biclustering algorithms to the “Ticket” task in the PISA 2012 CPS assessment, we evaluated the potential of biclustering algorithms in identifying and interpreting homogeneous biclusters from the process data. Compared with one-mode clustering algorithms, the biclustering methods could uncover clusters of individuals who are homogeneous on a subset of feature variables, holding promise for gaining fine-grained insights into students’ problem-solving behavior patterns. Empirical results revealed that specific subsets of features played a crucial role in identifying biclusters. Additionally, the study explored the utilization of biclustering on both the action sequence data and timing data, and the inclusion of time-based features enhanced the understanding of students’ action sequences and scores in the context of the analysis.

## 1. Introduction

Complex problem-solving (CPS) is a crucial skill in real-life settings that requires an individual to complete dynamic tasks by exploring and integrating information ([7]) and by using the obtained information to solve the task ([57]). To assess CPS skills, researchers have suggested using interactive tasks in simulated problem-solving scenarios ([13]). Some large-scale assessments, such as the Programme for the International Assessment of Adult Competencies (PIAAC; [43]) and the Programme for International Student Assessment (PISA; [44]), have utilized computer-based interactive tasks that mimic real-life problem-solving scenarios ([16]). Computer-based assessments offer the opportunity to collect and log a new source of behavioral data related to the process of problem-solving. Typically, this data includes the time-stamped sequence of actions recorded by the computer system as the individual pursues to solve the task ([4]; [43]). The new type of data is commonly referred to as process data or log file data ([18]; [61]). Compared with the traditional correct/incorrect response data on final outcome, process data can provide rich additional information that sheds light into the test takers’ problem-solving process, informing both assessment design and education ([24], [26]; [27], [28]).

Despite the richness of information, process data come often in a non-standard format, which can introduce complications to their analysis. Over the past decade, there has been a high demand for more innovative tools to handle this type of data ([64]). Feature extraction plays an important role in analyzing the process data, and there are two commonly used approaches. The first approach is theory-based feature extraction, which involves obtaining behavioral indicators based on established substantive theories ([22]; [68]). These methods often define task-specific features and require input and validation from domain experts. The second approach is data-driven feature extraction, which involves extracting features directly from the data without relying on substantive theories ([25]; [50]; [59], [60]; [62]). This approach utilizes statistical and machine learning techniques to identify patterns and relevant features in the process data. This presents a data-driven way for the identification of potentially meaningful sequential patterns that provide insights into strategies and cognitive processes. In addition to feature extraction from action sequences, methods for incorporating additional timing information have also been proposed, providing a more comprehensive investigation of examinees’ interactions ([61]; [63]).

Process features extracted from log files are often high-dimensional and sometimes lack direct interpretations. To uncover substantively meaningful latent structures that reflect individual differences in problem-solving, a popular approach from unsupervised machine learning is clustering algorithms. Clustering is commonly applied to the exploratory analysis of process data, aiming to identify groups that exhibit distinct interactions with the tasks ([26]; [50]; [52]; [63]). For instance, the widely used k-means method has been applied to analyze process data in PISA and PIAAC assessments, respectively, based on actions, the length of action sequences, and time spent on the item ([26]; [50]). Furthermore, researchers have also adopted more sophisticated or robust clustering methods. For example, the partitioning around medoids (PAMs) algorithm ([53]) was employed to explore the important role of multiple goals in the “Ticket” task in the PISA 2012 CPS test ([52]), and the *k-medoids* algorithm ([35]) was proposed for analyzing item CR551Q11 in the reading Rapa Nui unit in the PISA 2018 test ([25]). These more advanced clustering approaches aim to improve the accuracy and effectiveness of extracting meaningful patterns and insights from the process data.

Clustering techniques like hierarchical clustering and k-means clustering algorithms ([54]) are often rigid as they typically aim to find similarities considering all features of the subjects ([3]). The method of clustering data matrices applied in one direction is commonly referred to as one-mode clustering ([56]). However, when the number of features increases, the performance of these clustering methods may be poor: Intuitively, in the process data, certain behavioral patterns are commonly observed among a group of examinees under specific features. When the dimension of process features is high, different subsets of features may carry information on different kinds of individual differences in problem-solving patterns. In addition, there could be subsets of features that contain irrelevant information for clustering of examinees. Clustering based on individual differences in the full feature space would potentially mask these nuanced differences on specific dimensions. For instance, for the “Ticket” task in the PISA 2012 CPS test, through k-medoids clustering on the response time features, [47] ([47]) found the significant role of time-based features in characterizing students’ learning styles. However, the time-related features did not demonstrate an advantage as expected in the one-mode clustering method, k-means method when clustering the time-based features and action sequences simultaneously ([50]). In these cases, variable selection on the process features used for clustering could potentially improve the interpretability and within-cluster homogeneity in identified subgroups ([47]). To overcome the limitations of one-mode clustering, biclustering algorithms have been introduced and widely studied over the last two decades ([9]; [14]; [20]; [32]; [49]; [58]; [70]). Biclustering algorithms cluster the rows and columns of a data matrix simultaneously, allowing for more flexible and meaningful groupings of the data ([39]). Therefore, the biclustering algorithm is also referred to as two-mode clustering, signifying the approach of clustering data matrices in two directions ([56]). By considering both row and column clustering, biclustering methods can capture more specific and context-dependent patterns in the data, making them particularly useful for certain types of datasets and analysis tasks.

The aim of this study is to assess the performance of biclustering algorithms in analyzing process data and to understand the local homogeneity in the biclusters considering the importance of time-based features and action sequences simultaneously. In this paper, we conduct a comparative analysis of the capabilities of various biclustering algorithms in analyzing the process data. Specifically, we selected four popular and commonly used biclustering algorithms, including the binary inclusion-maximal biclustering algorithm (BIMAX; [49]), factor analysis for bicluster acquisition (FABIA; [15]; [30]; [46]) with Laplace prior, spectral biclustering ([42]), and [9]’s ([9]) biclustering algorithm (referred to as BCCC). We applied these biclustering algorithms along with the one-mode k-means algorithm to the “Ticket” task in the PISA 2012 CPS test. Through various evaluations, we identified the best-fitting clustering algorithm. The clustering results obtained from the best-fitting method were used to interpret timing and behavioral sequence patterns. Through comparison with the scores and results from one-mode clustering algorithms, we examine the utility of biclustering algorithms in analyzing the log file data. With these objectives, three specific research questions are as follows:RQ1.What are the representative biclusters from simultaneously clustering based on the
time-based features and action sequence features? Which important features have been utilized across different biclusters?RQ2.Is there any association between the resulting biclusters and students’ CPS performance?RQ3.What are the similarities and differences in biclusters and clusters, using the selected
biclustering and clustering algorithms?

The remainder of this paper is organized as follows. In Section 2, we introduce the “Ticket” task from the PISA 2012 CPS assessment under study, followed by a description of the participants and data sets utilized in this study. Additionally, in this section, we describe the biclustering algorithms and the evaluation metrics used for comparison. In Section 4, we discuss the testing configurations and present the results of the comparative analysis. Furthermore, we provide detailed interpretations of the biclustering patterns observed considering the timing data and action sequences. Finally, we conclude with a discussion that includes the implications and future directions.

## 2. Materials and Methods

### 2.1. Instrumentation

The 2012 PISA public dataset consists of 48 problem-solving items divided into 16 units, designed to assess participants’ cognitive skills in addressing real-world challenges through computer-based simulated scenarios ([44]). This study primarily focuses on the TICKETS task2 (CP038Q01) problem-solving item. More detailed information on this item can be found at the following site: https://www.oecd.org/pisa/test-2012/testquestions/question5/ (accessed on 1 September 2023). Figure 1 displays the screenshot of the opening page of this task, while Figure A1 provides screenshots of the pages illustrating the stages in solving the problem, as described by [50] ([50]). As depicted in Figure 1, this item provides students with a simulated problem-solving scenario involving ticket purchase. It requires them to determine the cheapest ticket option from the ticketing machine for a four-trip journey on the city subway. Furthermore, students have the option to purchase a concession fare as well. Therefore, this interactive question requires students to actively explore and collect the required information to make an informed decision ([44]). Therefore, the primary cognitive processes involved in this task are planning and execution ([12]; [21]; [44]; [47]; [50]).

To solve the problem, one possible solution is to select four individual concession tickets for the city subway, which costs 8 zeds. Alternatively, students can choose a daily concession ticket for the city subway, which costs 9 zeds. Both options are available as Figure A1 shows. Specifically, students have the option to select the *Cancel* button before making the final decision to *BUY*. By comparing the two alternative solutions, students can select the one with the lower cost. Therefore, correctly completing this task requires students to explore both paths and make a comparison. According to the scoring rubric, the item was scored polytomously with three score categories: 0, 1, and 2. Examinees who received a full score of 2 should purchase four individual city subway concession fares and compare their prices with the daily concession fare. A respondent who only tried one solution without comparison to the other option received a partial credit of 1. Students who made any other purchases or bought the wrong ticket received a score of 0 on this item.

### 2.2. Participants and Data Description

Similarly to [69] ([69]), the current study included respondents from the United States, Singapore, Austria, and Turkey. The dataset for the PISA 2012 log files related to the problem-solving item can be downloaded from the following website: http://www.oecd.org/pisa/pisaproducts/database-cbapisa2012.htm (accessed 1 September 2023). In the dataset, respondents with missing IDs and not-reached items were excluded from the analysis, resulting in a sample size of 3760. The missing responses were imputed using the full information maximum likelihood (FIML) estimation method ([69]). The raw process data comprised 43,338 rows and 11 variables as columns. An example log file for one student is provided in Figure A2.

The current study analyzed features generated from the raw process data following [50] ([50]). Specifically, four types of features were included: (1) time-based features, (2) single actions, (3) two actions coded together, and (4) four actions coded together as Table 1 in [50] ([50]) shows. According to the recommendations of [55] ([55]) and [50] ([50]), these features are considered theoretically significant in representing students’ problem-solving abilities, following the established scoring rules.

Furthermore, to provide a more precise interpretation of the clustering results, it is important to establish the connections between the features and the solutions. Identifying key and efficient sequences of actions is essential for addressing the problem at hand. Therefore, in this context, the most effective approach involves comparing the following two action sequences: *Start–city–subway–concession–individual–trip4* and *Start–city–subway–concession–daily*. It is worth noting that the *Cancel* action plays a crucial role. Therefore, features including *city_subway, concession, individual, trip_4, cancel, daily, S_city, city_concession, concession_daily, concession_individual, individual_trip4, daily_cancel, trip4_cancel, trip4_buy, city_con_ind_4, city_con_daily_cancel* are expected to appear frequently as high-frequency features.

Figure 2 displays the heatmap illustrating the values of features in the log files, arranged in ascending order of the total time spent on this task (*T_time*). Noticeable individual differences in the time features (*T_time, A_time, S_time*) can be observed. In terms of action-based features, subsequences such as *city_concession* and *city_subway* occurred with higher frequency compared to others. Certain features, such as *S_city*, were consistently used, while others, like *country_full* and *trip_1*, were utilized more rarely. Some features, like *Cancel*, exhibited variations among students similar to the trend of *T_time*. Moreover, the relationship between pairs of sequence features differed across subgroups of students. Specifically, only a subset of students simultaneously exhibited large *T_time* and *E_time*, similarly for *city_concession* and *Cancel*. This suggests that a biclustering approach may hold promise to identify features that move in unison among subgroups of students.

### 2.3. Methodology

#### 2.3.1. Biclustering Method

Over the past two decades, many biclustering methods have been proposed ([56]; [67]). For our study, we have chosen four widely recognized and commonly used biclustering algorithms: BIMAX, FABIA, Spectral, and BCCC, covering three search algorithms ([45]). These algorithms were selected due to their popularity and representation of diverse approaches in the field of biclustering methods ([6]; [11]; [29]; [39]; [45]; [49]; [67]).

The first method we selected was the BIMAX method, which dichotomizes the input data matrix and then recursively partitions it, searching for the submatrices consisting of entries that are all equal to one. This method employs a divide-and-conquer algorithm, making it a highly efficient method for detecting simple structures and combining their solutions into the solution for the original problem ([37]; [49]).

We then investigated two biclustering methods that utilize Distribution parameter identification algorithms ([39]): FABIA and Spectral. These methods assume statistical models associated with the structure of the biclusters. FABIA is a multiplicative model designed to extract linear dependencies between sample patterns and feature patterns ([15]; [30]; [46]). The Spectral Bicluster algorithm employs singular value decomposition to simultaneously cluster the features and samples ([36]). Additionally, the Spectral Bicluster algorithm preprocess the data matrix based on normalization.

Lastly, we selected the BCCC biclustering method, which utilizes the greedy approaches to find the best decision for a global optimal solution. It searches for submatrices in a standardized data matrix with a score lower than a specific threshold ([37]). The exhaustive advantages of the BCCC method enable efficient identification of temporally contiguous biclusters ([9]; [17]; [40]).

#### 2.3.2. Types of Biclusters

Assume that a total of *P* features are observed for *N* examinees in the test. Considering a data matrix $A={\left(a_{ip}\right)}_{1\leq i\leq N,1\leq p\leq P}$, where $a_{ip}$ denotes the *p*th feature of examinee *i*. The one-mode clustering algorithm can be applied separately to the rows (indexed as examinees) or the columns (indexed as features) of the data matrix. On the other hand, the biclustering algorithm clusters both the rows and columns simultaneously. Thus, the objective of biclustering techniques is to identify subgroups of examinees and subgroups of features by conducting concurrent clustering of both dimensions, rather than clustering them independently.

The biclustering algorithm aims to find subsets $B_{s}$ of the original data matrix *A*, for s=1,2,⋯,S, where *S* denotes the number of subsets. Each subset $B_{s}$ is called a bicluster denoted as $B_{s}={\left(b_{ij}\right)}_{|I_{s}\left|\times \right|J_{s}|}$, where $I_{s}\subseteq \left\{1,2,\cdots ,N\right\}$, $J_{s}\subseteq \left\{1,2,\cdots ,P\right\}$, and |D| denotes the cardinality of set *D*. Here, both examinees and features are allowed to overlap across biclusters. In other words, for $s_{1}\neq s_{2}$, the intersection of $I_{s_{1}}$ and $I_{s_{2}}$ can be nonempty, similarly for $J_{s}$.

Typically, the biclusters can be classified into three different types using various methods: constant bicluster, shifting bicluster, and scaling bicluster ([39]; [49]; [56]). Specifically, a constant bicluster is one where all elements within the bicluster are equal to the same constant value, making it the simplest structure. However, for real-valued matrices, it is appropriate to identify biclusters with coherent values. Therefore, the coherent biclustering algorithms often utilize parameters $\alpha_{i}$ and $\beta_{j}$ to adjust the rows (indexed by *i*) and columns (indexed by *j*) of the matrix. If $b_{ij}=\alpha_{i}+\beta_{j}$, a perfect shifting bicluster is discovered. If $b_{ij}=\alpha_{i}\times \beta_{j}$, a scaling bicluster is identified.

Figure 3 illustrates five examples of different types of biclusters, where rows represent students within a bicluster, and columns represent features within a bicluster. Figure 3a–c represent various constant biclusters. Figure 3a showcases a *perfect* constant bicluster where all values are identical. It’s important to note that such ideal biclusters are often masked by noise in real data ([39]). Figure 3b,c are also perfect biclusters, with either a subset of columns or rows consistently share the same values. On the other hand, Figure 3d depicts a shifting bicluster. In this type of bicluster, each row can be obtained by adding a constant to every row, while each column can be similarly obtained by adding a constant to each column. Similarly, Figure 3e illustrates a scaling bicluster. In a scaling bicluster, each row can be obtained by multiplying a constant to each of the rows, and each column can be obtained similarly by multiplying a constant to each of the columns.

Among the four selected biclustering algorithms, BIMAX aims to identify constant biclusters. It’s worth noting that BIMAX would sometimes dichotomize the data matrix, resulting in the constant biclusters that can also be considered as the scaling biclusters. Additionally, both FABIA and Spectral methods are designed to detect shifting biclusters, while the BCCC method focuses on constant biclusters. However, it’s important to know that in the analysis of PISA tasks, the type of bicluster represented in Figure 3b is highly unlikely to occur because an examinee’s behavior typically varies across different features.

#### 2.3.3. Evaluation Metrics

To compare the performances of biclustering algorithms, several internal measures can be employed to evaluate the identified biclusters ([8]; [39]). The first statistic is the within-bicluster variance (VAR; [23]),
(1)$$\mathrm{VAR}\left(B_{s}\right)=\frac{1}{|I_{s}\left|\times \right|J_{s}|}\sum_{i\in I_{s}}\sum_{j\in J_{s}}{\left(b_{ij}-b_{I_{s}J_{s}}\right)}^{2},$$
where $b_{ij}$ denotes the element (i,j) in the bicluster $B_{s}$ for $i\in I_{s}$, $j\in J_{s}$, and $b_{I_{s}J_{s}}$ denotes the mean of the elements in the bicluster $B_{s}$. Usually, the criterion VAR would be effective in measuring within-bicluster homogeneity for constant biclusters but fail on more complex patterns.

As an alternative method to evaluate within-bicluster similarity, the mean squared residue (MSR; [9]) is employed as a measure of the coherence over the rows and columns within the bicluster. The specific expression of MSR is given as follows,
(2)$$\mathrm{MSR}\left(B_{s}\right)=\frac{1}{|I_{s}\left|\times \right|J_{s}|}\sum_{i\in I_{s}}\sum_{j\in J_{s}}{\left(b_{ij}-b_{iJ_{s}}-b_{I_{s}j}+b_{I_{s}J_{s}}\right)}^{2},$$
where $b_{iJ_{s}}$ and $b_{I_{s}j}$ denote the mean of the row *i* and the column *j* in bicluster *s*, repectively.

Another measure aims to find the coherence of the biclusters based on the normalized pattern, called virtual error (VE; [1]; [10]). The specific expression is given as follows,
(3)$$\mathrm{VE}\left(B_{s}\right)=\frac{1}{|I_{s}\left|\times \right|J_{s}|}\sum_{i\in I_{s}}\sum_{j\in J_{s}}\left|{\hat{b}}_{ij}-{\hat{\rho }}_{j}\right|,$$
where the (i,j) normalized element in bicluster *s* is
$${\hat{b}}_{ij}=\frac{b_{ij}-\mu_{i}^{\left(s\right)}}{\sigma_{i}^{\left(s\right)}},$$
$\mu_{i}^{\left(s\right)}$ represents the mean of row *i* in bicluster *s*, and $\sigma_{i}^{\left(s\right)}$ denotes the standard deviation of row *i* in bicluster *s*. Let ${\hat{\rho }}_{j}=\sum_{i\in I_{s}}{\hat{b}}_{ij}/\left|I_{s}\right|$ denote the normalized pattern over rows in bicluster *s*. [48] ([48]) pointed out that virtual error can effectively capture both scaling and shifting patterns. [38] ([38]) pointed out that VE is a suitable internal measure to compare different biclustering results.

The specific problem addressed by biclustering algorithms can now be defined. Given a data matrix, our goal is to identify a set of biclusters $B_{s}$, in such a way that each bicluster $B_{s}$ exhibits certain defined characteristics of homogeneity. The exact criteria for homogeneity may differ between different approaches ([39]). For instance, the similarity score, referred to as MSR, is one such criterion.

## 3. Data Analysis

Here, we outline the procedures taken in the current study to apply biclustering to the analysis of CPS log data. The procedures can be broken into 6 steps:

### 3.1. Procedure 1: Specification of Features

Before implementing biclustering, the first step is to specify the data matrix of features in the CPS data. In this article, we employed the features defined in [50] ([50]) shown in Table 1 in [50] ([50]) and Figure 2. These 36 features, consisting of 32 action-based features and 4 time-based features, were derived in a theory-driven manner, where actions and subsequences in the log data with potential relationship with the measured construct of CPS were preserved. Such theory-driven feature engineering aids the interpretability of in subsequent clustering.

### 3.2. Procedure 2: Pre-Processing of Input

In our analysis, we chose not to standardize the features used in both clustering and biclustering to preserve the original scale of the data. As depicted in Figure 2, the values of time-based features are notably larger than those of the action sequences. Standardization, in this case, would bring the action sequence and time-based features onto the same scale. However, preserving the original scale of the data, which captured presence/absence/frequency of actions and time spent, is crucial for meaningful interpretation within the context of complex problem-solving assessment.

It’s worth noting that some biclustering algorithms perform feature preprocessing within the algorithm. For instance, the BIMAX algorithm dichotomizes the input data matrix and subsequently seeks submatrices, followed by a post-processing step to restore the original data matrix. The Spectral and FABIA methods employ standardization based on a standard normal distribution. Conversely, the BCCC method does not apply any feature transformation during preprocessing.

### 3.3. Procedure 3: Configuration Testing and Selection of Biclustering Methods

In this procedure, the parameter settings for the BIMAX, FABIA, Spectral and BCCC algorithms were optimally set to the default values based on the aforementioned internal measures. For the BIMAX and FABIA algorithms, the number of possible biclusters varied from 2 to 30, as presented in Table A1 and Table A2. The aforementioned evaluation metrics including VAR, MSR, and VE, were employed to select the optimal number of clusters for BIMAX and FABIA algorithms. Further, evaluation metrics were employed to compare BIMAX, FABIA, Spectral and BCCC algorithms as well as k-means clustering.

### 3.4. Procedure 4: Interpret the Homogeneous Biclusters and Contribution of Features

First, under biclustering, subsets of features were utilized in different biclusters resembling sparsity. Therefore, the overall contribution of a feature was evaluated based on the number of times the feature was included across all the biclusters. Furthermore, to explain each bicluster, we examined the features included in the bicluster, as well as the magnitude on these features for examinees within the bicluster. A *homogeneous* bicluster contains examinees who are similarly high (or low) on the included features. In the analysis of the PISA CPS data, we picked several representative biclusters that exhibited high homogeneity on the selected features, to illustrate how the identified biclusters could be interpreted.

### 3.5. Procedure 5: Track the Association between the Biclusters and Scores

Based on the selected optimal biclustering solution, the distribution of scores was analyzed within each bicluster. Further, through the level of scores, biclusters were classified into four different types, including clusters that corresponded to scores of 0, 1, 2, or a mix of different scores.

### 3.6. Procedure 6: Comparison between the Biclustering and One-Mode Clustering

Finally, to understand the similarity and differences in grouping of examinees based on biclustering versus k-means clustering, we cross-tabulated the examinees’ memberships in biclusters with their memberships in the clusters identified based on k-means clustering.

## 4. Results

This section discusses the findings from applying biclustering to the PISA CPS data. BIMAX, FABIA, Spectral, and BCCC were applied using two R packages: *biclust* ([33]), and *fabia* ([31]). Additionally, we compared these biclustering algorithms with clustering based on k-means. The k-means clustering was carried out using the *“k-means”* function in the R package *stats* ([51]).

### 4.1. Biclusters Identified from PISA CPS Ticket Item

#### 4.1.1. Testing Configurations

To facilitate the comparison among these algorithms, we commenced by selecting the optimal number of clusters for each algorithm. As Table A1 and Table A2 showed, the criteria VAR, MSR, and VE consistently indicated the optimal solutions for the BIMAX and FABIA algorithms. Specifically, the twelve-bicluster solution for BIMAX and the ten-bicluster solution for FABIA demonstrated the lowest values across all three criteria. These findings strongly suggest that participants should be clustered into twelve and ten clusters, respectively, when utilizing the BIMAX and FABIA algorithms. For the Spectral algorithm, we determined the number of eigenvalues considered to find biclusters based on our internal measures and ultimately chose 2 as the optimal number. On the other hand, for the BCCC algorithm, we set the target number of biclusters to be found as 100. Figure 4 visually illustrates the results of our analysis, indicating that the Spectral algorithm identified 12 clusters, while the BCCC algorithm detected a total of 22 clusters.

For the k-means method, we considered a maximum of 30 clusters to be found. We employed two different criteria to determine the optimal number of clusters. The first criterion was the average silhouette width, which is a commonly used measure. We employed the NbClust method to determine the optimal number of clusters for the data set using the *“fviz_nbclust”* function in the R package *factoextra* ([34]). We calculated the average silhouette width for cluster numbers ranging from 1 to 30 and depicted the relationship in Figure 5. Notably, the two-cluster solution exhibited the highest silhouette width, indicating that two clusters would be the optimal choice. The second criterion utilized was the evaluation of VAR, MSR, and VE. Table A3 presents the mean values of VAR for different cluster numbers, which aligned with the results depicted in Figure 5, suggesting that the optimal number of clusters was also two according to VAR. However, when considering the criteria MSR and VE, different solutions emerged. MSR identified a total of 20 clusters as the optimal configuration, while VE suggested a higher number of clusters, specifically 29.

Based on the optimal clusters selected from the analysis, the comparison results across different biclustering and clustering methods are presented in Table 1. Regarding VAR, the BIMAX method was identified as the best-performing method. However, when considering the criteria of MSR and VE, the BCCC method consistently outperformed the other clustering methods. This suggests that the BIMAX method’s ability to discover high-quality biclusters may be limited since it yielded the highest values for the more appropriate measures MSR and VE. Additionally, both the Spectral and FABIA methods encountered similar challenges when considering the MSR and VE criteria. Their factorization-based approach primarily focuses on detecting specific types of bicluster structures, which restricts their ability to find diverse biclusters. Therefore, the BCCC method was suggested as the optimal clustering method in this study.

Furthermore, Table 1 demonstrates that all the biclustering methods exhibited smaller values of VAR within clusters compared to the k-means method. This observation suggests that biclustering techniques tend to generate more homogeneous structures compared to one-mode clustering approaches.

#### 4.1.2. Contribution of Features

The primary aim of biclustering is to cluster both rows and columns of a data matrix simultaneously to discover homogeneous submatrices. With this approach, these techniques can potentially identify overlapping submatrices ([45]). Consistent with results of [45] ([45]), among the four selected methods, varying degrees of overlap were observed in the features. However, the BCCC and FABIA biclustering methods did not reveal any overlap among examinees, while the other two methods identified a significant amount of overlap.

Figure 6 shows the frequency of features across clusters for different biclustering methods. As for the Spectral method, each of the 36 features was used four times, hence the results are not presented in Figure 6. The BIMAX method utilized only 11 out of 36 features to identify biclusters, while the FABIA and BCCC methods employed 34 and 36 features, respectively. Furthermore, in the case of the k-means method, all 36 features were utilized to find each cluster. This indicates that the k-means method considered the entirety of the available features without any feature selection or exclusion. Consequently, each feature contributed to the clustering process, and no specific feature was deemed less important or excluded from consideration.

From Figure 6, it could be concluded that the feature *city_con_daily_cancel* played a significant role in the biclustering algorithms, which was consistent with the findings in [50] ([50]). Moreover, in line with the phantom items defined by experts in [69] ([69]), certain action sequences such as *concession*, *Cancel*, and *city_concession* were also identified as important features for identifying coherent biclusters. Overall, these results suggest that biclustering algorithms strive to strike a balance between interpretability and data mining outcomes.

Note that these biclustering algorithms could select not only frequently used features for students but also those features that students would not typically utilize. For instance, in the case of the BCCC algorithm depicted in Figure 6, the feature *full_fare* was identified as one of the top nine frequently occurring features. However, this item specifically asked students to purchase a concession ticket, making it a distractor. Out of the total of 3760 students, 3192 students did not use this option at all, and only 11 students attempted to use *full_fare* button more than five times. This finding aligns with the nature of biclustering algorithms, which tend to prioritize the identification of coherent biclusters. The same observation applies to the *country_concession* feature in the BIMAX algorithm and the *country_trains* feature in the FABIA algorithm, respectively.

Furthermore, it is worth highlighting that the time-related features, including *T_time*, *A_time*, *S_time*, and *E_time*, were consistently selected as important features for achieving coherence among the biclusters in Figure 6. This suggests that there exists a similarity in terms of time-based patterns within certain coherent clusters. The significance of time-related features in classification is to be expected, since time intuitively plays a crucial role in problem-solving and learning dynamics. This finding is also consistent with the observations in [66] ([66]), where the presence of rapid learners in complex problem-solving scenarios was identified. Therefore, the inclusion of time-related features further enhances the understanding of students’ behaviors and performance in the context of the analysis.

### 4.2. Explanation of Representative Biclusters

Based on the results presented in Section 4.1, the BCCC method was selected as the optimal biclustering algorithm. Consequently, in the following subsections, we specifically focus on conducting an in-depth analysis based on the clustering results obtained from the BCCC method.

Figure 7 and Figure 8 depict the specific performances of features for different clusters based on the results of the BCCC method. Additionally, Table A4 and Table A5 in the appendix offer the feature averages for each bicluster based on the outcomes of the BCCC method. Specifically, Figure 7 provides a summary of all the features, while Figure 8 presents the summary of the selected features within a specific cluster in the BCCC method.

First, comparing Figure 7 and Figure 8, we can observe that the features that were excluded in each bicluster typically exhibited larger within-cluster heterogeneity. For example, for time-based features, only *A_time* was preserved in Cluster 9, and other time-based features presented a high variability, as indicated by bars of various colors in Figure 7. In comparison, all the time-based features were preserved in Cluster 10, where individuals within the cluster were highly similar on these features. Similar trends were observed for Clusters 9 and 10 on action-based features. For example, *daily* and *country_trains* were excluded in Cluster 9.

When a feature was included in a bicluster, the average within-cluster magnitude on the feature could either be high or low. For example, in Clusters 9 and 10, features *city_subway* and *concession* exhibited homogeneity with high average values, while features *trip_1* and *country_full* exhibited homogeneity with low values, suggesting that individuals within these biclusters seldom performed these two subsequences.

Comparing Clusters 9 and 10, it can be observed that on the included features, the magnitudes across these two clusters were relatively similar. In particular, examinees in both biclusters showed homogeneity on performing subsequences necessary for full credit, i.e., *Start–city–subway–concession–individual–trip4*, *Start–city–subway–concession–daily*, and *Cancel*. The difference between the two biclusters primarily lied in the included feature set. For Cluster 10, all action and time features were included, while Cluster 9 excluded three time features as well as action features that pertained to subsequences less relevant to the correct response (e.g., ones involving clicking *daily* and *country*). Individuals in Cluster 9 demonstrated greater heterogeneity on these less relevant actions, despite that they performed the necessary steps for completing the task after all.

### 4.3. Relationship between Biclusters and Performances

Figure 9 presents the number of examinees in each cluster under the BCCC method. Firstly, we ranked the clusters based on the number of examinees in each cluster in decreasing order. As depicted in Figure 9, Cluster 1 contains the highest number of examinees, with a total of 1021 individuals, whereas Cluster 22 consists of only 2 examinees, which is the lowest among all clusters.

Figure 10 depicts a visual representation of the distribution for scores across various levels within each cluster, based on the BCCC method. Specifically, one observes distinct patterns in the score distribution across clusters. In particular, all the examinees in Clusters 10 and 12 answered the question correctly and achieved the full score of 2. Additionally, almost all examinees in Cluster 9 obtained the full score of 2. On the other hand, Clusters 2, 6, and 19 contained examinees who received partial credit with a score of 1, suggesting these examinees were able to purchase the correct type of ticket, without implementing the price comparison. In contrast, all the examinees in Clusters 21 and 22 and a high proportion of examinees in Cluster 3 displayed the lowest scores 0, which may indicate failure to execute the task based on the explicit requirements.

The remaining clusters (such as Clusters 1, 4, 5, and 20) exhibited a mixture of scores among the students as Figure 10 shows. This suggests some degree of homogeneity in the actions or timing patterns among the students within these clusters, despite receiving different scores. This observation can be attributed to the biclustering algorithm, which primarily focuses on students’ problem-solving strategies from a behavioral perspective. This PISA ticket item was designed to assess exploring and understanding, requiring students to adopt a targeted exploration strategy and to synthesize the information about prices of different alternatives ([44]). Thus, students with different levels of proficiency on exploring and understanding may produce diverse results, even if they employ similar actions ([66]).

#### 4.3.1. Biclusters with Full Score

Clusters with a higher proportion of perfect scores, a full score of 2, demonstrated mostly homogeneous structures in their action sequences. As Figure 8 showed, in Clusters 9, 10, and 12, the essential actions in *Start–city–subway–concession–individual–trip4*, *Start–city–subway–concession–daily*, and *Cancel* were frequently utilized, while the irrelevant buttons were seldom chosen. From a behavioral perspective, this indicates that these students did compare the prices of individual tickets for four trips and a daily ticket, then finally made a correct decision to buy the cheaper one. Furthermore, examinees in Cluster 9 made more attempts to use the *Cancel* button compared to the other two clusters. Examinees in Cluster 12 used feature *city_concession* more frequently compared to the other two clusters.

#### 4.3.2. Biclusters with Partial Score

Furthermore, clusters with a higher proportion of partial scores also exhibited homogeneous strategies. For instance, examinees in Cluster 2 utilized the key action sequence *Start–city–subway–concession–daily*, while rarely used other buttons. This indicates that these students purchased a daily ticket with minimum additional explorations, including comparing it to the prices of individual tickets. Similarly, examinees in Clusters 6 and 19 frequently employed the key action sequence *Start–city–subway–concession–individual–trip4*, while rarely using other buttons. This implies that these students bought individual tickets for four trips but did not compare them to the prices of a daily ticket. However, examinees in Cluster 19 used the *Cancel* button more frequently, whereas examinees in Cluster 6 rarely used it.

#### 4.3.3. Biclusters with a Score of 0

Examinees in Cluster 3 exhibited relatively homogeneous structures in the action sequence, *Start–city–subway–concession*, but then became divided on the subsequent choices of *daily* or *individual*. They additionally lacked consecutive actions, where key subsequences including *city_con_ind_4* and *city_con_daily_cancel* were rarely chosen. As a result, the majority of examinees in Cluster 3 obtained a score of 0. Furthermore, examinees in Cluster 22 demonstrated homogeneous structures by repeatedly attempting the features *full_fare*, *trip_5*, and *full_individual* for a long time, which also resulted in all of them obtaining a score of 0. In summary, examinees who scored 0 displayed more diversified strategies in the problem-solving process compared to individuals who answered correctly.

#### 4.3.4. Biclusters with a Mixture of Scores

Interestingly, we have also observed some similarities in clusters that include a mixture of scores. For instance, in Cluster 4, examinees made multiple attempts on the action sequence *city_con_daily_cancel*, particularly focusing on the features *city_subway*, *city_concession*, and *Cancel*. Remarkably, many of these students still managed to provide correct responses despite their numerous attempts. Similar findings can be observed in Cluster 5, where students made multiple attempts to compare the action sequences *Start–city–subway–concession–individual–trip4* and *Start–city–subway–concession–daily* for a longer action time.

### 4.4. Comparison of the Biclustering and Clustering Solutions

#### 4.4.1. Homogeneity in Time-Based Features

Compared to one-mode k-means clustering, in addition to its effectiveness in clustering examinees, biclustering did indeed highlight the significant role of time-related features, as expected. Specifically, it is evident that time-based features, including features *T_time*, *A_time*, *S_time*, *E_time*, did play a crucial role in the BCCC method, as colors have segregated into distinct layers among different clusters, particularly concerning the total time of action. The results presented in Figure 8 offer a clearer visualization of this.

Despite the fact that examinees in Cluster 9, 10 and 12, achieved perfect scores, their usage of time varied. Examinees in Cluster 10 showed homogeneity in terms of total time, action time, and relatively low start and end times. However, examinees in Cluster 12 exhibited similarity in time features except for the end time, while examinees in Cluster 9 displayed similarity only in the action time. This suggests the presence of homogeneity in time-based features within different clusters.

Additionally, there were differences in the time utilization between Cluster 6 and 19, with examinees in Cluster 19 taking longer total and action times to solve the problem compared to those in Cluster 6. To some extent, this suggests that examinees in Cluster 6 made decisions more decisively than those in Cluster 19.

#### 4.4.2. Degree of Similarity of Biclustering and Clustering

In Table 2, cross-tabulations are presented between student memberships derived from biclusters and categories generated by the k-means method given two clusters. For instance, among the 134 students in Bicluster 9, 74 were assigned to the first k-means cluster, while 60 were assigned to the second k-means cluster. Therefore, Table 2 presents the relationship between two groups of biclusters and k-means solution under two clusters. For clusters with a higher proportion of perfect scores, Clusters 9, 10 and 12 were nearly equally represented in both categories KM2-Cluster1 and KM2-Cluster2. However, for clusters with a higher proportion of partial scores, all the students in Clusters 2 and 6 presented in the second category KM2-Cluster2 and Cluster 19 were mostly present in the second category KM2-Cluster2.

Figure 11 provided the feature heatmap of biclusters 9, 10 and 12, crossed with the results from *k*-means given two clusters. Cluster 10 showed homogeneity considering all the features as k-means method did. However, within Cluster 10, the sequence patterns of KM2-Cluster1 and KM2-Cluster2 were very similar, although slight differences were found on selecting timing features. The same trend showed up in Clusters 9 and 12. Further, the biclustering algorithm differentiated the three clusters considering the subset of features. Cluster 9 removed several features from action sequences for biclustering, and Cluster 12 excluded a time feature.

Based on results in Table 2, there were very few data points in Clusters 2, 6 and 19 belong to KM2-Cluster1. Thus, Figure 12 only provides the heatmap of Cluster 2, 6 and 19 which can nearly denote the comparison with k-means given two clusters. For Clusters 2 and 6, all the features have been employed in classification. There are clear differences between the two biclusters, especially in action features. However, KM2-Cluster2 classified them into one category.

## 5. Discussion

This study investigated the utilization of biclustering algorithms in the context of process data. Four different well-established biclustering algorithms were considered, along with the one-mode clustering algorithm based on k-means. To evaluate their capabilities of identifying the homogeneous groups, we analyzed a dataset consisting of 3760 students who participated in the “Ticket” task of the PISA 2012 CPS test. Then we applied the four biclustering algorithms using a set of well-defined features. To comprehensively evaluate the four biclustering and compare with the one-mode k-means clustering method, we consider three different evaluation metrics: variance, mean squared residue, and virtual error metrics. The results indicated that all the biclustering methods clearly outperformed the one-mode clustering approach when using the variance metric. This suggests that biclustering techniques tend to produce more homogeneous structures. Furthermore, the BCCC biclustering algorithm consistently outperformed the other methods when considering the mean squared residue and virtual error metrics. From the perspective of evaluation metrics, we find that biclustering algorithms are useful in achieving homogeneity on subsets of features.

Further, focusing on the individual performances of the four biclustering algorithms, the BCCC method exhibited high homogeneity levels in generating some small biclusters. To comprehensively understand respondents’ problem-solving competence and leverage the rich information contained in the process data, we examined the achievement data and process data simultaneously under the BCCC method. Specifically, for examinees who obtained a full score, their action sequence patterns revealed a high level of homogeneity in the biclustering results. For examinees who obtained partial credit, their action sequence patterns depicted a moderate level of homogeneity. However, for examinees who had incorrect responses, the biclusters demonstrated diverse action sequence patterns, such as frequent attempts on the wrong buttons or being stuck at certain steps. Moreover, we have also observed that students may produce diverse results even though they employed similar exploration strategies ([66]). That is because the BCCC biclustering algorithm identified some homogenous structures across different scores, suggesting that similar problem-solving strategies were employed by students despite variations in their overall performance.

In conclusion, the biclustering technique is a powerful tool for discovering local patterns in process data analysis. By using features extracted from the process data established in prior literature, we demonstrated how to utilize biclustering algorithms to recognize and interpret homogeneous biclusters in the PISA 2012 CPS test. Our study highlights the promising potential of biclustering algorithms in analyzing process data to identify prototypical problem-solving patterns.

### 5.1. Biclustering and Clustering Methods

The key distinction of the biclustering algorithms from one-mode clustering is that they utilize only a subset of features instead of all features for clustering each observation, striking a balance between interpretability and capturing the individual differences across all features in the observed data. Our findings indicated that certain features played a crucial role.

First, through extraction of relevant features from the selected biclustering approaches, the study gave a broader understanding of contributions of features. On one hand, features like *city_con_daily_cancel* were consistently included in multiple biclusters, aligning with previous findings from data mining techniques ([50]). On the other hand, certain action sequences such as *concession*, *Cancel*, and *city_concession* were frequently employed, in agreement with results obtained from phantom items defined by experts ([69]). Additionally, the biclustering algorithms successfully identified homogeneous patterns in some less frequently used features.

In addition, as expected, the biclustering algorithms succeeded to capture the individual differences on time-based features, which outperformed the one-mode k-means method ([50]). Specifically, the time-related features in classification were found to be more relevant in certain clusters rather than being coherent across all biclusters.

For classification of examinees, the biclusters refined the categorization of the clusters under the k-means method. On one hand, the study identified cross-similarities between biclusters and clusters. On the other hand, the study differentiated biclusters from the categorization by k-means.

### 5.2. Implications

Our study has several implications for measurement. First, the information stored in the log files largely depends on the design of the user interface. Our results revealed that contribution differed across features with high homogeneity and some buttons did not differentiate examinees. Such evaluation will provide a deeper insight into the design of simulation tasks. For example, how challenging or difficult these irrelevant buttons should be set. Second, previous researches found the crucial role of time-based features. However, time-based features were not successfully identified in k-means clustering. The current study provided a more comprehensive view of the time-based feature and action sequences. Homogeneity could only be found in certain groups of students implying a universal skill among all examinees. Such information might support modeling and explanation of response time in evaluating students’ proficiency.

### 5.3. Considerations in Interpreting Biclusters

In one-mode clustering, like hierarchical clustering and k-means clustering algorithms, all features are considered when evaluating the dissimilarity of observations (e.g., examinees) within a cluster. Therefore, in the selection of the number of clusters, using within-cluster variability (e.g., sum of squares) as the selection criteria tends to favor solutions with a large number of clusters. However, in biclustering, the number of features included in each (bi)cluster of examinees is different—depending on the subset of examinees in a given bicluster. The biclustering algorithms also identify subsets of features that act in unison for that subset of examinees. As a consequence, breaking examinees into a larger number of smaller biclusters, which may increase the number of nuisance features included in each bicluster, will not necessarily lead to a decrease in within-bicluster variability across the included features. As the feature set changes in biclustering, we note that the meaning of within-cluster variability in biclustering inherently differs from that of one-mode clustering, which readers might be more familiar with. At the same time, the concept of between-cluster variability, which evaluates how dissimilar are observations from different clusters, is no longer straightforward in the biclustering case, because the biclusters can overlap in examinees, and the selected features differ across different biclusters. In interpreting biclustering solutions that are selected based on within-bicluster variability measures, rather than focusing on the partitioning of the examinees as in one-mode clustering, researchers should treat it as an algorithm that provides information on (potentially overlapping) subsets of examinees, who share similarities on certain features. The biclustering solutions selected based on measures such as MSR and VE are intended to identify subsets that are homogeneous, therefore facilitating the interpretation of each bicluster by making it more “pure” on the included dimensions. At the same time, we note that there are limitations to evaluation indices based on within-bicluster variability, as it does not consider solution parsimony: For instance, in the empirical example, the optimal solution based on VE and MSR contains 22 biclusters, which might be far more than needed for capturing cognitively meaningful individual differences on the ticket item. We leave it to future research to develop evaluation indices or methods for selecting the optimal number of biclusters that build in parsimony. In the meantime, rather than interpreting all identified biclusters which can be very taxing, substantive researchers interpreting the biclustering solutions may prioritize the interpretation of biclusters that either (1) contain a sufficient number of examinees of interest or (2) are linked to key variables of interest, e.g., task outcome, proficiency, or demographic background.

It is also worth mentioning how biclustering connects to other types of clustering-related methods. The first is *soft clustering*. Methods such as *k*-means and hierarchical clustering are *hard clustering* algorithms, in that each examinee gets deterministically assigned to one class. Soft clustering, for instance via a probabilistic mixture model (e.g., [41]), still assumes that each examinee comes from a single class but characterizes class membership as the probability that an examinee stems from each class, rather than a single class label. In this sense, biclustering could also be seen as a hard two-mode clustering algorithm, in that for each bicluster, binary decisions are made for whether each examinee and feature is included. Another related type of probabilistic model is *mixed-membership models* (e.g., [2]), which, similar to biclustering, relaxes the assumption that each examinee originates from a single class. When repeated measures are available for an examinee (e.g., multiple items or features), these models assume that the observations for the same examinee can stem from different latent classes, with the class probabilities characterized by an examinee-specific class proportion vector. Well-known examples include the grade of membership model ([65]) and latent Dirichlet allocation ([5]). Different from biclustering, most mixed-membership models do not simultaneously group features into latent classes. However, recent extensions, such as the dimension-grouped mixed-membership model ([19]), additionally allow the grouping of feature dimensions, where the features from the same group act in unison in the latent class they originate from for a given examinee. Future research may consider the application of these models to the analysis of log data and compare it to biclustering.

### 5.4. Limitations and Future Directions

The current study has limitations and can be extended in several ways. Firstly, time-based features in process data are not limited to total time, action time, and end time. Some researchers have discovered there are other important time-based features in the process data. For example, [47] ([47]) established item-level response time data, while [12] ([12]) discovered the elapsed time of completing a state transition. Future studies can further consider incorporating more precise time-based features in the biclustering algorithms to effectively identify homogeneous structures. Additionally, in this paper, we only employed four commonly used biclustering algorithms, but many other interesting biclustering algorithms have been developed. In the future, we could explore and compare more biclustering algorithms to gain deeper insights into their performances. Third, the validation of biclustering algorithms is also one of the most important issues that need to be addressed.

It is also worth emphasizing the limited scope of substantive conclusions that can be drawn from the current exploratory study. The current study aims to introduce an exploratory process data analysis approach based on biclustering, for uncovering types of problem-solving patterns. Although the exploratory findings could serve as a data-driven guide to uncover meaning individual differences in problem-solving processes and proficiency, interpretations on the cognitive processes or strategies underlying observed patterns are speculative. For instance, some buttons that were frequently used within certain clusters, e.g., clicking *cancel*, could be attributed to a multitude of reasons, ranging from familiarizing oneself with the task’s structure via exploring various routes, to an interrupted problem-solving process featured by frequent restarts. Therefore, the exploratory findings and speculations based on observed sequential/time patterns should not be construed as establishing theoretical models for cognitive processes, which require follow-up confirmatory analyses and formal scientific inferences.

## Figures and Tables

**Figure 1 jintelligence-12-00010-f001:**
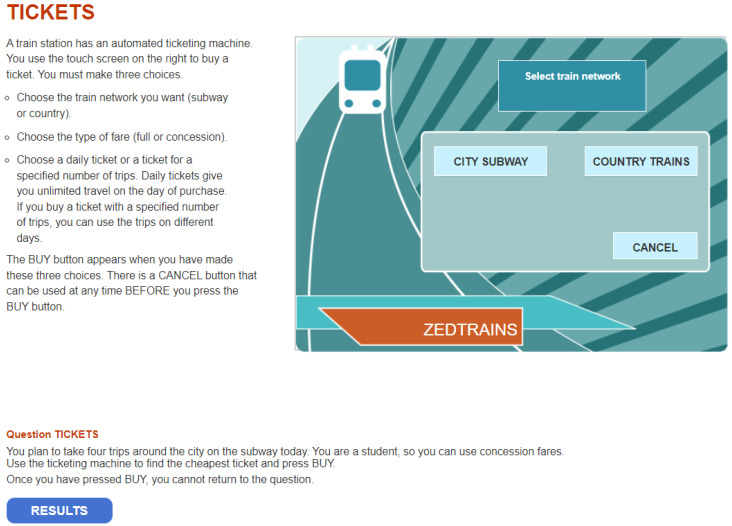
The screenshot of the opening page of PISA 2012 problem-solving question TICKETS task2 (CP038Q01).

**Figure 2 jintelligence-12-00010-f002:**
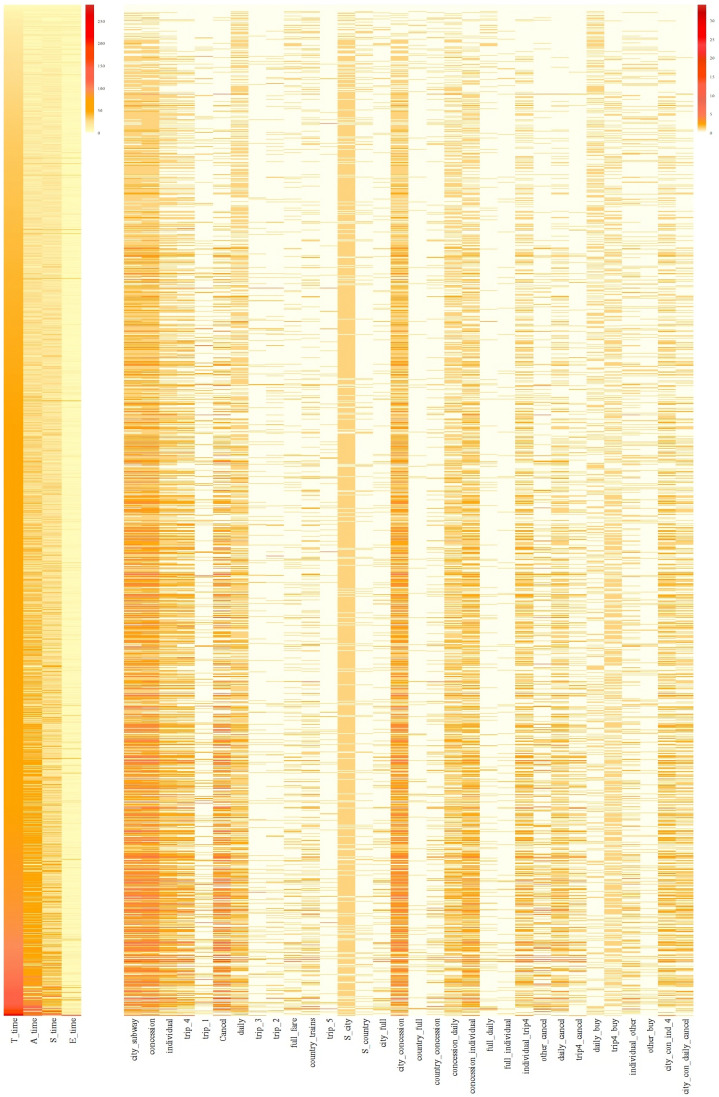
Heatmap of the log file for PISA 2012 problem-solving question TICKETS task2 (CP038Q01).

**Figure 3 jintelligence-12-00010-f003:**
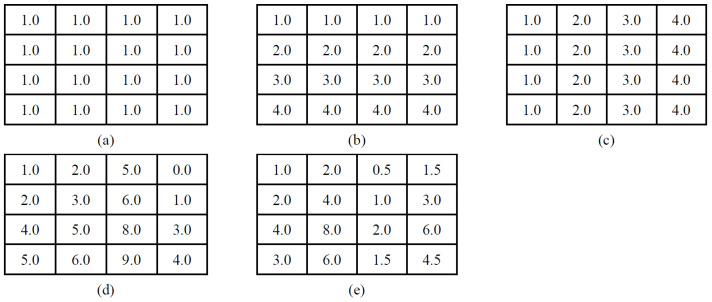
Examples of different types of biclusters following [39] ([39]). (**a**) Constant bicluster, (**b**) constant rows, (**c**) constant columns, (**d**) shifting bicluster, and (**e**) scaling bicluster.

**Figure 4 jintelligence-12-00010-f004:**
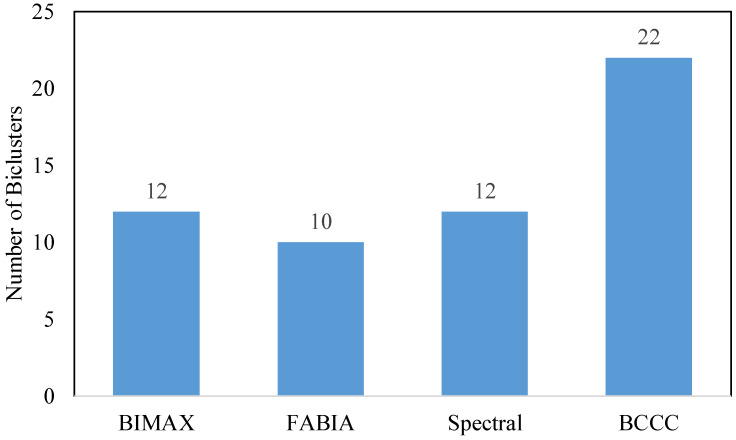
The number of biclusters for different biclustering methods.

**Figure 5 jintelligence-12-00010-f005:**
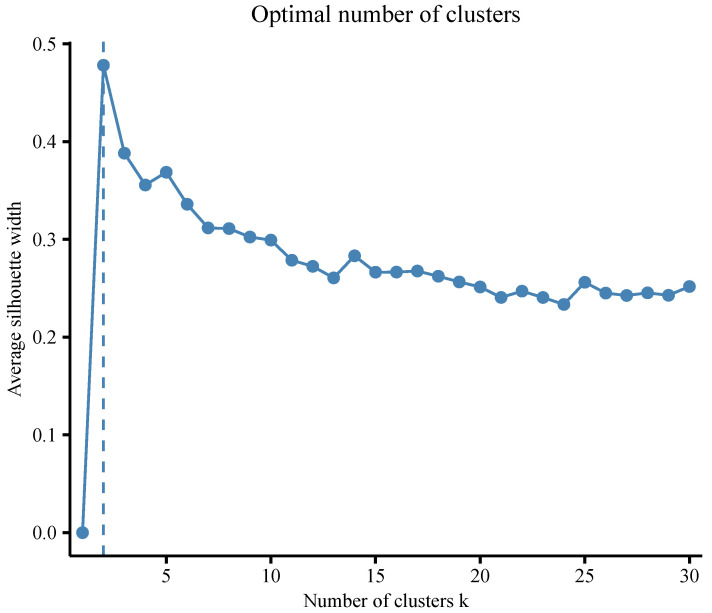
The optimal number of clusters by the average silhouette width for the k-means clustering.

**Figure 6 jintelligence-12-00010-f006:**
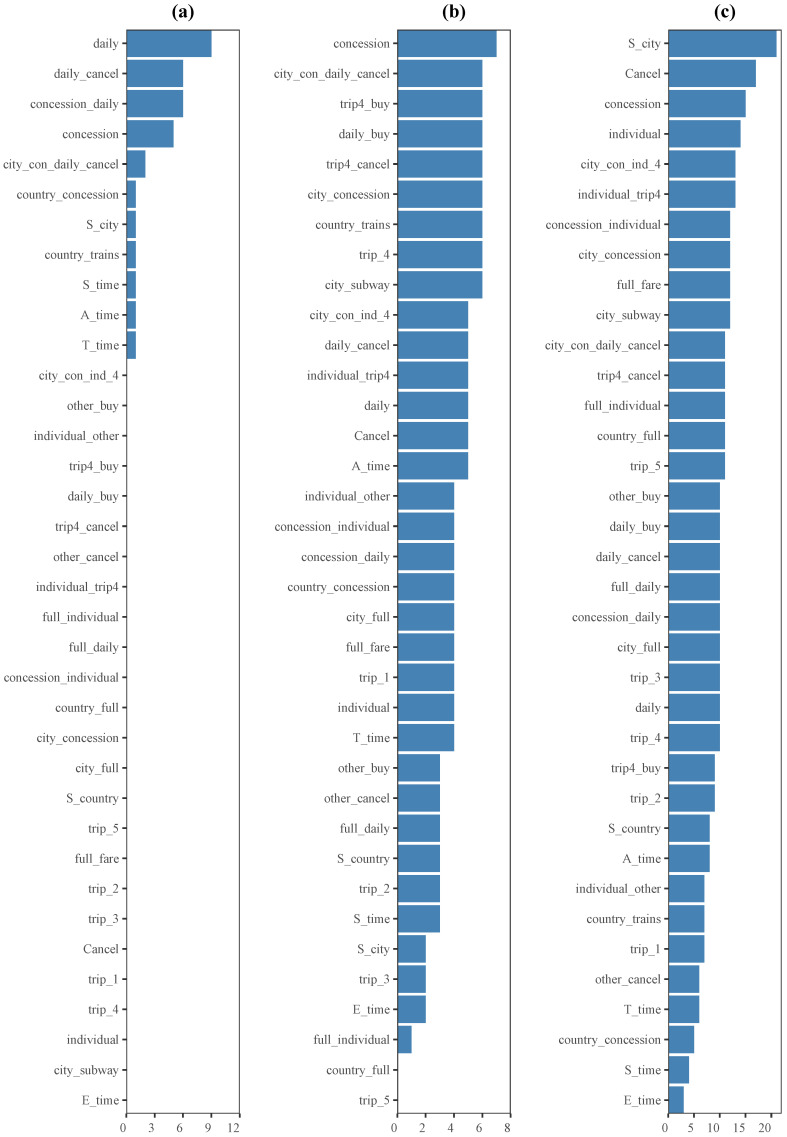
The frequency of features across the clusters under different biclustering methods. Figures (**a**–**c**) represent the results of the BIMAX, FABIA and BCCC methods, respectively. The frequency of features across the clusters under different biclustering method.

**Figure 7 jintelligence-12-00010-f007:**
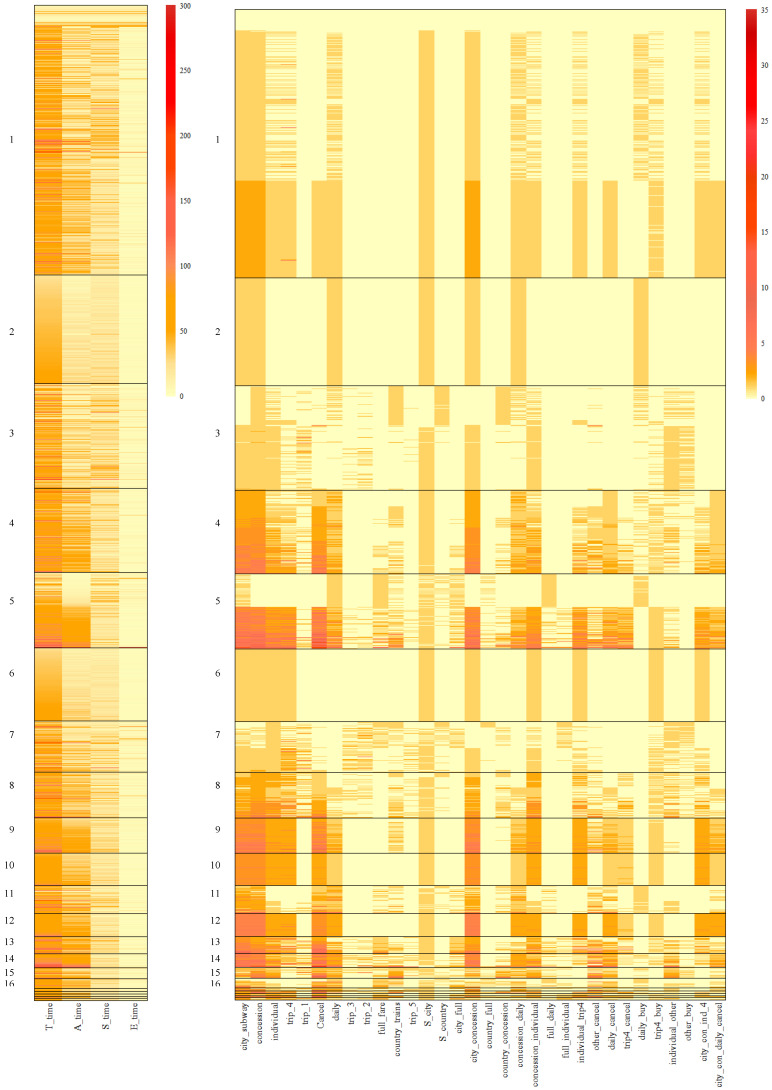
Heatmap of all the features within the clusters in the BCCC method.

**Figure 8 jintelligence-12-00010-f008:**
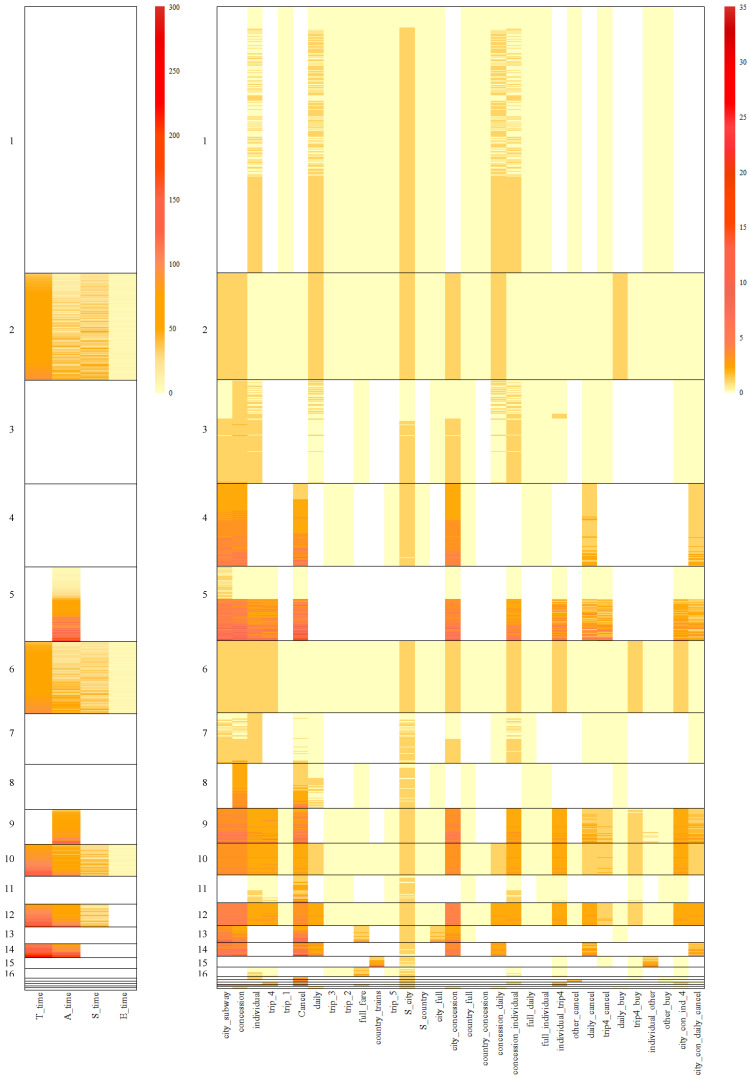
Heatmap of the selected features within clusters in the BCCC method. Features in white were not selected in a bicluster. Features in light yellow were selected with within-cluster mean around 0.

**Figure 9 jintelligence-12-00010-f009:**
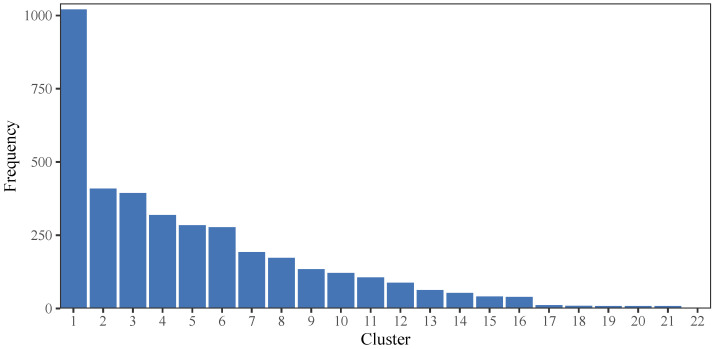
The number of examinees in each cluster considering BCCC method.

**Figure 10 jintelligence-12-00010-f010:**
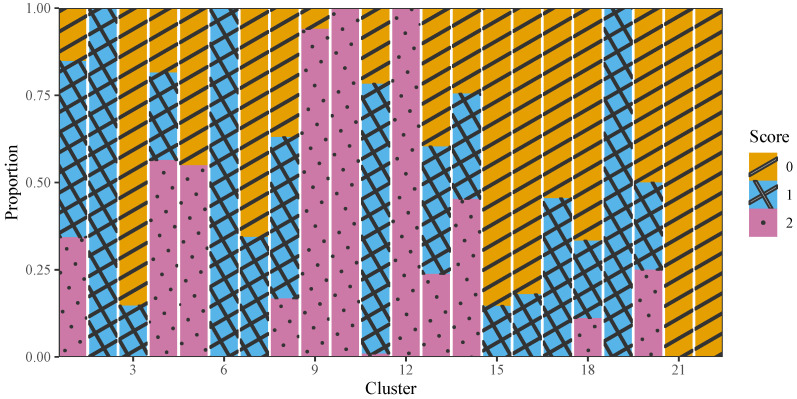
The proportion of different scores in each cluster considering BCCC method.

**Figure 11 jintelligence-12-00010-f011:**
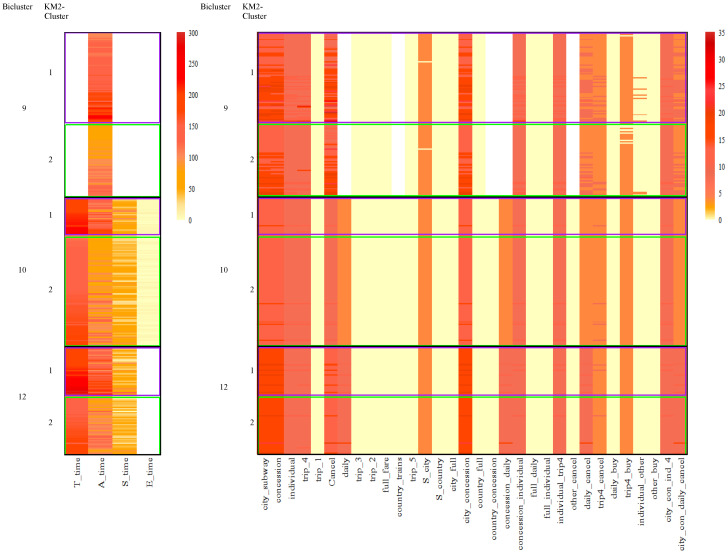
Heatmap of Cluster 9, 10 and 12 cross the results from *k*-means given two clusters. From top to bottom, the black rectangles represent biclusters 9, 10 and 12 respectively. Within each black rectangle, the purple rectangle represents individuals classified based on k-means into KM2-Cluster1, while the green rectangle represents those in KM2-Cluster2.

**Figure 12 jintelligence-12-00010-f012:**
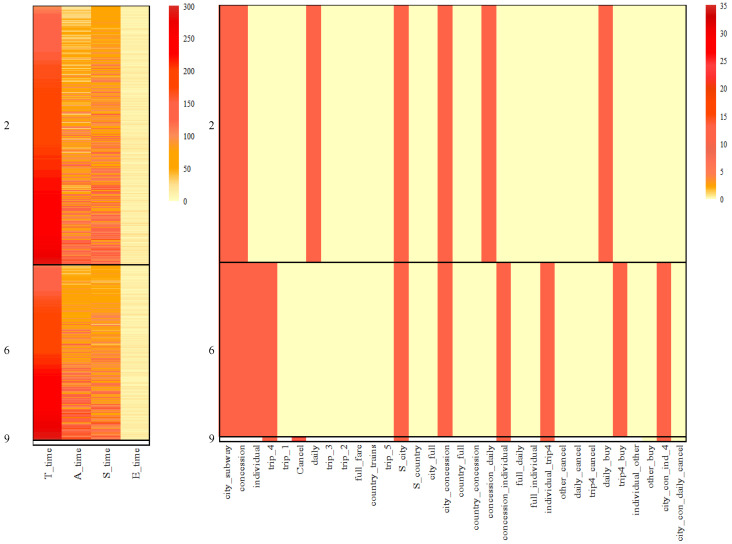
Heatmap of Cluster 2, 6 and 19 cross the results from *k*-means given two clusters. From top to bottom, the black rectangle represents bicluster 2, 6 and 19.

**Table 1 jintelligence-12-00010-t001:** Comparison of means of different criteria across different methods.

	KM2	KM29	KM20	BIMAX	FABIA	Spectral	BCCC
VAR	201.316	454.621	498.897	**86.832**	93.002	149.118	146.941
MSR	26.961	18.173	16.266	11.195	23.977	25.337	**10.502**
VE	1.562	1.122	1.123	2.151	1.304	1.398	**0.794**

*Notes*: KM2 denotes k-means method given two clusters, and the same to KM29 and KM20. VAR denotes the variance of the cluster, MSR denotes the mean squared residue, and VE denotes the virtual error.

**Table 2 jintelligence-12-00010-t002:** Relationship between two groups of biclusters and k-means method given two clusters.

Bicluster	KM2-Cluster1	KM2-Cluster2	Bicluster	KM2-Cluster1	KM2-Cluster2
9	74	60	2	0	409
10	31	90	6	0	277
12	40	48	19	2	6

*Notes*: KM2 denotes k-means method given two clusters, KM2-Cluster1 denotes the first k-means clusters and the same to KM2-Cluster2.

## Data Availability

The data analyzed in the current study is publicly available from the PISA public database, which can be found at https://www.oecd.org/pisa/pisaproducts/database-cbapisa2012.htm (accessed on 1 September 2023).

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
