# Peer review of "Biclustering of Log Data: Insights from a Computer-Based Complex Problem Solving Assessment"

_jintelligence, 2024, doi:10.3390/jintelligence12010010_

Round 1

Reviewer 1 Report

Comments and Suggestions for Authors

Please find my comments in the attached file.

Author Response

Thank you so much for the constructive feedback on the manuscript. Please find attached our point-by-point responses. 

Reviewer 2 Report

Comments and Suggestions for Authors

The article “Biclustering of Log Data: Insights from a Computer-based Complex Problem-Solving Assessment” describes biclustering methods in application to data analysis in complex problem-solving research. The article is quite well-written and presents new and complex material in a way that is easy and interesting to read, which clearly indicates the author's knowledge in the area. PISA tasks, used by the authors to perform analysis on, are well-known among researchers in the area and thus help to dive into the methodology right away without much additional reading. This also helps with the delivery of the main message of the article. Yet, this main message is not clear enough, which settles the main problem with this article. It is generally difficult to identify its genre.

On the one hand, the authors could have been aiming to contribute to complex problem-solving methodologies. In this case, the general idea might have been to show that the proposed analysis plan (biclustering) provides advantages when analysing complex problem-solving strategies. Yet, in this case, the article did not pay enough attention to complex problems as the research subject. Specifically, the authors said nothing about the problem of identifying and analysing strategies and their psychological mechanisms. Those questions have been thoroughly discussed in the existing literature; e.g., PISA tasks, used in the current article, are known as minimal complex systems, and for them are described such strategies as VOTAT (“Vary One Thing at a Time”) and others. Those strategies were well represented in the works of Joachim Funke, and at least one of his works was known to authors, as it has been referenced in the Introduction section of the article (Funke, 2001). Those strategies and the discussions surrounding them play an important role in complex problem-solving research, so it might be useful for any new methodology proposed for this research area to address them. For example, in the current article, relying on the existing strategy analysis methodology could be used to justify the selection of subsets of features for biclustering. The article, however, simply divided those features into temporary and effective ones. The lack of a theoretical model that describes the key parameters of problem-solving strategies and their relation to log data does not allow for a connection between the results of biclustering and any understanding of the psychological mechanisms behind the observed behaviour, making the results interesting in general but of limited usefulness for psychological research. In addition, when analysing logs, it is important to understand that what is saved in the log files largely depends on the design of the interface of the complex problem being solved. For example, repeatedly clicking “cancel” and switching between “city subway” and “country trains” can have various reasons, from the desire to become familiar with the structure of the task in detail to being the result of decreased working memory and difficulties with concentration. Moreover, identifying a large number of clusters rather complicates the task of identifying generalised strategies. The authors say nothing about the psychological mechanisms behind their results, so it is possible that it was not the focus of the study, but in this case, it is questionable whether or not this work fits into the scope of the Journal of Intelligence, as the intended readers would probably be more interested in the application of the actual analysis strategies applied to the complex problem-solving research.

On the other hand, the main aim of the study could have been to justify the greater possibilities of biclustering in comparison with more traditional clustering methods (e.g., k-means clustering, included in this work) and to compare different methods of biclustering (BIMAX, FABIA, etc.), with the complex problems being more of the material than a focus point. In this case, the attention paid to complex problems seems redundant in the part where the authors meaningfully analyse the selected clusters, interpreting the behavioural patterns of the individuals composing them. And, on the other hand, the article fails to provide some sort of checklist or a strategy for a future researcher who might want to use biclustering in their studies after reading this article; the authors do not suggest any particular method of biclustering, nor do they explain how those methods could be chosen in the first place. In this particular research, the BCCC methods were described as the most fitting, but this result does not provide any insight for future research.

To put it simply, the article should be more focused either on the statistical or on the complex problem-solving part of itself. In the current state, it aims for both but fails to provide a compelling reading for both the reader interested in CPS and the reader interested in introducing new methods in their psychological or cognitive research. The main conclusion of the article seems to be the line, “The biclustering technique is a powerful tool for discovering local patterns in process data analysis,” which might be slightly irrelevant to the audience of the journal, with the scope mainly in psychology but not in mathematics.

As for the minor observations, Figure 7 utilises a red-green-blue colour scheme, which seems to be very visual-illusion-inducing due to the use of complimentary colours. E.g., on some screens I was clearly seeing a thin one-pixel green line between blue and red in the 9th column, while on the other screens that line disappeared. While not being colourblind myself, I also assume this colour scheme might be a problem for a colourblind reader, so I suggest changing the colours somehow to be more distinctive.

Sometimes the authors also used a strangely formalised approach to the description of the results. For example, on page 11, the authors state that Cluster 22 in the BCCC method consisted of only 2 examinees (line 353). On page 14, when describing the strategies, the authors claim that in cluster 22, the examinees used homogenous strategies, which “resulted in most of them obtaining a score of 0” (lines 401–402). Figure 7 is fully coloured red, suggesting that not “most”, but all the examinees in the cluster got a 0 score, and it is also quite difficult to apply the wording “most of them” to the group of 2.

Overall, while I think the article is very interesting and could be useful for the readers, my opinion is that to fit into the scope of the Journal of Intelligence, the authors should specify the main goal of their study and aim either for a more in-depth psychological approach or for a more applied statistics for cognitive sciences approach, with the appropriate changes in the discussion and conclusion sections of the article.

Author Response

(The authors gave the same response as above.)

Round 2

Reviewer 1 Report

Comments and Suggestions for Authors

I appreciate the authors' attention to my previous comments, which have been addressed well. I have two additional minor comments:

1. Line 563 page 20: Should "select" be "selecting"?

2. Line 567 on page 20: Do you mean that very few data points in Clusters 2, 6, and 19 belong to KM2-Cluster1?

Author Response

Thank you again for the constructive feedback throughout the revision of the manuscript! We have addressed the minor comments in the revision. Please find attached our responses.

Reviewer 2 Report

Comments and Suggestions for Authors

As I have already done a relatively thorough review of the previous version of the article, this review would be much shorter and deviate from the regular scheme, as the strong points of the article “Biclustering of Log Data: Insights from a Computer-based Complex Problem Solving Assessment” remain the same. As for the changes introduced by the authors, most of them directly address the criticism. Those efforts are strongly appreciated. The authors improved the clarity and accuracy of their statements, providing an overall better experience for the reader. The methodological part of the article was expanded significantly and made clearer for a casual reader, and the expanded Data Analysis section is helpful for a possible reproduction of the results as well as applying similar methods to one`s own data sets. The Discussion and Limitation sections further explain the aim of the study and the possible application of the results for future research. Due to this, I can agree that the article in its current state can be published in a peer-reviewed journal. Yet, however, my main concern remains the same: I am not quite sure the article matches the scope of the journal it was uploaded to (Journal of Intelligence), as it does add almost nothing new to intelligence research or the understanding of Complex Problem Solving as a topic in cognitive sciences or psychology. Mostly, the authors used the tasks from PISA as an illustration of the methods they were inserted in applying. Due to those concerns, I strongly suggest the Editorial Board look into the article by themselves to see if it matches the current journal or should be redirected to another MDPI journal with a more matching aim and scope. Aside from them, I see the article as interesting and completed research that might find its reader and generally contributes to enhancing our knowledge about biclustering methods applied to log data analysis.

Author Response

Thank you so much for the constructive feedback and thoughtful comments throughout the revision of the manuscript! Please find attached our response letter.

Round 3

Reviewer 1 Report

Comments and Suggestions for Authors

I don't have additional comments.